# Ovarian Cancer in a Northern Italian Province and the Multidisciplinary Team

**DOI:** 10.3390/cancers15010299

**Published:** 2022-12-31

**Authors:** Lucia Mangone, Francesco Marinelli, Isabella Bisceglia, Maria Barbara Braghiroli, Valentina Mastrofilippo, Loredana Cerullo, Carlotta Pellegri, Alessandro Zambelli, Lorenzo Aguzzoli, Vincenzo Dario Mandato

**Affiliations:** 1Epidemiology Unit, Azienda Unità Sanitaria Locale—IRCCS di Reggio Emilia, 42122 Reggio Emilia, Italy; 2Unit of Obstetrics and Gynaecology, Azienda Unità Sanitaria Locale—IRCCS di Reggio Emilia, 42122 Reggio Emilia, Italy; 3Quality Office, Azienda Unità Sanitaria Locale—IRCCS di Reggio Emilia, 42122 Reggio Emilia, Italy

**Keywords:** ovarian cancer, stage, multidisciplinary team, recurrence, disease free, death

## Abstract

**Simple Summary:**

Ovarian cancer is one of the most aggressive tumors in the world due to its clinical, biological, and molecular complexity. It is considered a “silent killer” due to the lack of specific symptoms that delay diagnosis. A multidisciplinary approach can improve the prognosis both in terms of recurrences and death, especially in the first 24 months after diagnosis, by changing the type of treatment, reducing recurrences and mortality, or increasing survival.

**Abstract:**

Ovarian cancer represents one of the most aggressive female cancers in the world, remaining a tumor with high lethality. This study aims to present how a multidisciplinary team (MDT) approach can improve the prognosis in terms of recurrence and death of patients. In total, 448 ovarian cancer cases registered in an Italian Cancer Registry between 2012 and 2020 were included. Information on age, morphology, stage, and treatment was collected. Recurrence and death rates were reported 1 and 2 years after diagnosis, comparing MDT vs. non-MDT approaches. Ninety-three percent had microscopic confirmation, and most showed cystic-mucinous morphology. In total, 50% were older than 65 years old. The distribution by stage was 17.6%, 4%, 44.9%, and 32.6% for stages I, II, III, and IV, respectively. The women followed by the MDT were 24.1%. Disease-free survival 1-year post-diagnosis, recurrences, recurrences-deaths, and deaths were 67.5%, 14.5%, 8.4%, and 9.6%, respectively, better than the non-MDT group (46.2%, 13.2%, 20.8 %, and 19.8%, respectively) (*p* < 0.01). The same positive results were confirmed two years after diagnosis, particularly for stages III and IV. Albeit small numbers, the study confirms a better prognosis for women managed by MDT with fewer recurrences and deaths, especially within the first 24 months of diagnosis.

## 1. Introduction

Ovarian cancer represents the eighth most frequently diagnosed tumor and the eighth most lethal cancer in women, leading to almost 200,000 deaths annually worldwide [1]. Roughly 5179 new cases of ovarian cancers (2.8% of the total of female cancers) are diagnosed in Italy, with regrettably 3336 deaths (4.2% of total deaths) [2]. Ovarian cancer usually affects 55–65-year-old women [3], and in 86–90% of cases, it presents epithelial morphologies [4]. Over the decades, the therapeutic options for the treatment of ovarian cancers have been improved significantly through the advancement of surgical techniques and the availability of novel effective drugs able to extend the life expectancy of patients [5]. However, due to its clinical, biological and molecular complexity, ovarian cancer remains one of the most challenging tumors to manage. Accordingly, 5-year survival remains very low (42.7%) in Italy, even for cases diagnosed in recent years (2010–2014) [6]. The incidence trend in Italy appears to be decreasing in the period 2003–2018 (APC = −0.8%; −1.2; −0.4), and mortality appears to be slightly decreasing (APC = −0.1%; −0.5%; 0.3) [7]. Ovarian cancer is considered a “silent killer” due to the lack of specific symptoms that delay diagnosis. It is often misdiagnosed as a common intestinal discomfort, back pain, cystitis, etc., and advanced disease is found when the right late diagnosis is achieved [8,9]. Several studies and scientific societies have advocated the centralization of ovarian cancer patients, and a multidisciplinary approach is essential both in the decision-making strategies and during surgical procedures [10,11,12], but unfortunately, this recommendation is still disregarded [13]. A multidisciplinary team (MDT) could improve the outcome of the cancer patient by changing the type of treatment, reducing recurrences and mortality, or increasing survival [14]. Furthermore, an MDT approach appears to be able to reduce recurrences in breast cancer [15,16,17] and rectal cancer [18] or reduce lung cancer mortality [19]. Meaningful results were also obtained on head and neck [20,21,22] and esophagus [23] tumors, as well as cholangiocarcinoma [24]. For ovarian cancer, the MDT could increase survival by 40% [25,26] and reduce recurrence [27] significantly in specialized centers [28,29].

This study aims to present recent advantages regarding disease-free survival (DFS) and death of ovarian cancer patients, emphasizing the benefits of an MDT.

## 2. Materials and Methods

Data from this population-based cohort came from the Reggio Emilia Cancer Registry (RE-CR), approved by the provincial Ethics Committee of Reggio Emilia (Protocol no. 2014/0019740 of 4 August 2014.). The leading information sources of the RE-CR are anatomic pathology reports, hospital discharge records, and mortality data, integrated with laboratory tests, diagnostic reports, and information from general practitioners. The RE-CR covers a population of 532,000 inhabitants and is considered a high-quality CR thanks to the fact that its data are up to date (the incidence data extend to the end of 2020), with a high percentage of microscopic confirmation (91.5% for ovarian cancer) and a low rate of DCO (Death Certificate Only, <0.1%) [30]. A clinical/instrumental diagnosis includes cases without microscopic confirmation but which have been confirmed by instrumental diagnosis (abdominal ultrasound, abdominal CT scan, etc.) and validated by clinical information.

Ovarian cancer cases were defined based on the International Classification of Diseases for Oncology, Third Edition (ICD-O-3) as topography C56 [31]. The study included information on all ovarian cancer patients diagnosed from 2012 to 2020, divided into MDT and non-MDT. The MDT of cancer patients has been implemented in Reggio Emilia since 2015 and discusses all cases of ovarian cancer that are diagnosed in the province. The team that participates in the MDT includes gynecologists, pathologists, radiologists, oncologists, support psychologists, and palliative care doctors. 

At the MDT, all diagnosed cases that will undergo cytoreduction, cases that will undergo neoadjuvant chemotherapy, and patients with recurrences are discussed. The team meets once a week. To compare the outcome of the MDT approach with the general trend of ovarian cancers in the province of Reggio Emilia, the incidence and mortality trends for the 2001–2020 period are also reported. Information on stage (TNM 8th edition) [32] and information on surgery and chemotherapy were collected by consulting the medical records in the hospital. 

Some characteristics of the patients were presented using descriptive statistics also stratified by group status (MDT and non-MDT). Fisher’s exact and χ^2^-tests were performed to evaluate differences between groups, and 1-year and 2-year outcomes were defined as follows: disease-free for the duration of the observation period, a recurrence occurred without mortality, a recurrence followed by death, or death (without recurrence) occurred within the designated time frame. Differences in recurrence and mortality outcomes between MDT and non-MDT patients were evaluated using Fisher’s exact test and stratified by stage to address the imbalances in disease severity between groups. We also performed logistic regression analyses for recurrence and mortality while accounting for possible predictors: MDT status, age at diagnosis, tumor stage, morphology, surgery, and chemotherapy. The follow-up of the study is updated to 31 December 2021; for this reason, we excluded the year 2020 from the 2-year outcome. The standardized incidence and mortality rate for the last 20 years (2001–2020) was calculated. Population estimates, which were used to derive rates, are represented by the general population of the Province of Reggio Emilia recorded on 1 January of each year. Incidence and incidence-based mortality rates were adjusted to the 2013 European standard population and calculated per 100,000 person-years. Analyses were performed using STATA 16.1 software. In this study, we reported 95% confidence intervals (CI) and defined a *p*-value < 0.05 as statistically significant. Trends over time were analyzed by calculating the annual percent change (APC) in age-standardized rates using Joinpoint Regression.

## 3. Results

From 2012 to 2020, 448 ovarian cancers were collected from Reggio Emilia Cancer Registry. The main characteristics of the patients are described in Table 1. In most cases (50.5%), the tumors concern the age group 65+, have a microscopic confirmation (92.7%), the most frequent morphology is represented by cystic mucinous and serous neoplasms (58.7%), have an advanced tumor (Stage III and IV: 77.5%), had surgery and chemotherapy (approximately 65%). The MDT took over 24.1% of cases. The distribution of tumors by year and stage is shown in Table 2: 17.6% of tumors are in stage I, 4.0% in stage II, 44.9% in stage III, and 32.6% in stage IV. Notably, three-quarters of tumors were in advanced or metastatic status at the onset of the disease. To note, there has been a small increase in stage I associated with a slight decrease in stage IV over the years.

The cases managed by the MDT (Table 3) are younger, have a higher percentage of histological confirmations, and more frequently present cysto- mucinous-serous morphologies, stage I, and less metastatic forms. In total, 75% of the cases managed by MDT underwent surgery, and 82.4% underwent chemotherapy. The number of recurrences is 25% in the MDT and 26.8% in the non-MDT group. Stratifying by stage (Table 4), there are no significant variations in DFS and recurrence in 1 year and 2 years after diagnosis in stages I and II. Stage III presents, after 1 year, higher percentages of DFS (54.1% vs. 25.3%) and lower recurrence-death values (10.8% vs. 40.4%) (*p* < 0.01). The same results are observed 2 years after diagnosis in the MDT group: DFS 92.3% vs. 45.6% and recurrence 7.7% vs. 27.9%; similarly, fewer recurrence/deaths and deaths were observed (*p* < 0.01).

Even for stage IV, the DFS 1 year after diagnosis has higher values (60% vs. 38.8%) and fewer deaths than non-MDT (26.7% vs. 55.1%). At 2 years, the values become statistically significant in the MDT group both for DFS (84.2% vs. 62.5%) and for mortality (15.8% vs. 32.5%; *p* < 0.20). 

After adjusting for possible predictors, logistic regression analyses showed a decreased mortality at 2 years, OR = 0.31 (95% CI 0.09; 1.0) but not at 1 year, OR = 0.95 (95% CI 0.40; 2.24) among MDT and non-MDT patients. However, recurrence was significantly lower at 2 years in the former group, OR = 0.18 (95% CI 0.05; 0.68) and also at 1 year, OR = 0.49 (95% CI 0.19; 1.27) (data not shown). Ovarian cancer incidence and mortality data (Figure 1) show stable trends from 2001 to 2020, with APC = 0.2 (95% CI -0.9; 1.3) for incidence and APC = 0 (95% CI -2; 2) for mortality.

**Figure 1 cancers-15-00299-f001:**
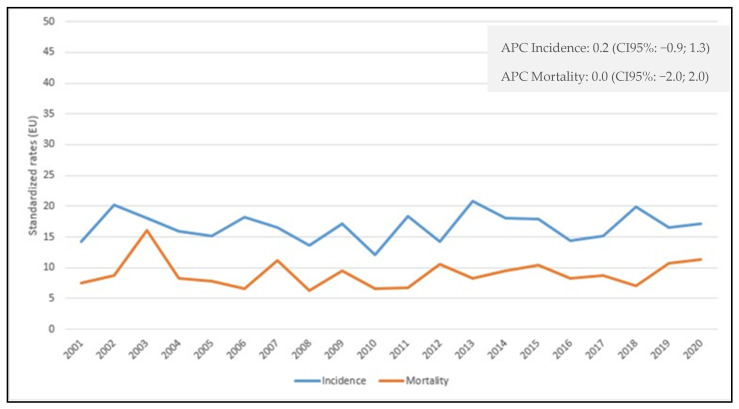
Age-standardized incidence and mortality rates per 100,000 person-years in the Province of Reggio Emilia in the period 2001–2020.

**Table 1 cancers-15-00299-t001:** Distribution of cases by age, diagnosis, stage, and treatment. Years 2012–2018.

	n	%
Overall	448	
Age at diagnosis		
<50	77	17.2
50–65	145	32.3
65+	226	50.5
Method of diagnosis		
Histological	369	82.4
Cytological	46	10.3
Clinical/instrumental	33	7.3
Morphology		
Cystic, mucinous, and serous neoplasms	263	58.7
Adenomas and adenocarcinomas	68	15.2
Neoplasms, NOS	60	13.4
Epithelial neoplasms	34	7.6
Germ cell neoplasms	9	2.0
Sarcoma	8	1.8
Granulosa cell tumor and Sertoli–Leydig cell tumor	6	1.3
Stage		
I	79	17.6
II	18	4.0
III	201	44.9
IV	146	32.6
Unknown	4	0.9
MDT		
Yes	108	24.1
No	340	75.9
Surgery		
Yes	290	64.7
No	158	35.3
Chemotherapy		
Yes	293	65.4
No	136	30.4
Unknown	19	4.2

**Table 2 cancers-15-00299-t002:** Distribution of cases by stage and year of diagnosis.

	Year of Diagnosis									
Stage	2012	2013	2014	2015	2016	2017	2018	2019	2020	Total
	n (%)	n (%)	n (%)	n (%)	n (%)	n (%)	n (%)	n (%)	n (%)	n (%)
I	3 (7.7)	9 (16.1)	6 (11.3)	13 (24.1)	7 (16.3)	7 (14.9)	14 (23.3)	10 (21.7)	10 (20.0)	79 (17.6)
II	1 (2.6)	3 (5.4)	3 (5.7)	5 (9.3)	1 (2.3)	0 (0.0)	0 (0.0)	1 (2.2)	4 (8.0)	18 (4.0)
III	18 (46.2)	24 (42.8)	29 (54.7)	20 (37.0)	19 (44.2)	21 (44.7)	28 (46.7)	20 (43.5)	22 (44.0)	201 (44.9)
IV	17 (43.5)	20 (35.7)	15 (28.3)	16 (29.6)	16 (37.2)	16 (34.0)	18 (30.0)	15 (32.6)	13 (26.0)	146 (32.6)
Unknown	0 (0.0)	0 (0.0)	0 (0.0)	0 (0.0)	0 (0.0)	3 (6.4)	0 (0.0)	0 (0.0)	1 (2.0)	4 (0.9)
Total	39 (100)	56 (100)	53 (100)	54 (100)	43 (100)	47 (100)	60 (100)	46(100)	50 (100)	448(100)

**Table 3 cancers-15-00299-t003:** Distribution of cases by age, diagnosis, stage, and treatment for MDT patients.

	MDT				
	Yes (*n* = 108)		No (*n* = 340)		*p*-Value
	n	%	n	%	
Age at diagnosis					0.14
<50	25	23.2	52	15.3	
50–65	36	33.3	109	32.1	
65+	47	43.5	179	52.6	
Method of diagnosis					<0.01
Histological	101	93.5	268	78.8	
Cytological	7	6.5	39	11.5	
Clinical/instrumental	0	0.0	33	9.7	
Morphology					<0.01
Cystic, mucinous, and serous neoplasms	77	71.3	186	54.7	
Adenomas and adenocarcinomas	21	19.5	47	13.8	
Neoplasms, NOS	1	0.9	59	17.4	
Epithelial neoplasms	4	3.7	30	8.8	
Germ cell neoplasms	1	0.9	8	2.3	
Sarcoma	1	0.9	7	2.1	
Granulosa cell tumor and Sertoli–Leydig cell tumor	3	2.8	3	0.9	
Stage					<0.05
I	28	25.9	51	15.0	
II	4	3.7	14	4.1	
III	52	48.2	149	43.8	
IV	24	22.2	122	35.9	
Unknown	0	0.0	4	1.2	
Surgery					<0.05
Yes	81	75.0	209	38.5	
No	27	25.0	131	61.5	
Chemotherapy					<0.01
Yes	89	82.4	204	60.0	
No	19	17.6	117	34.4	
Unknown	0	0.0	19	5.6	
Recurrence					0.86
Yes	27	25.0	91	26.8	
No	78	72.2	237	69.7	
Unknown	3	2.8	12	3.5	

*p*-value was calculated with Fisher’s exact test or χ^2^-test as appropriated.

**Table 4 cancers-15-00299-t004:** Distribution of outcomes by stage for MDT and non-MDT patients, one and two years after diagnosis.

	1 Year		*p*-Value	2 Year		*p*-Value
	MDTYes (%)	MDT No (%)		MDT Yes (%)	MDT No (%)	
Stage 1			0.64			0.53
Disease-free	88.9	86.0		92.2	89.6	
Recurrence	7.4	4.0		3.9	6.3	
Recurrence ->Died	3.7	6.0		3.9	0.0	
Died	0.0	4.0		0.0	4.2	
Stage 2			0.62			0.72
Disease-free	75.0	81.8		100	81.8	
Recurrence	25.0	9.1		0.0	9.1	
Recurrence -> Died	0.0	0.0		0.0	0.0	
Died	0.0	9.1		0.0	9.1	
Stage 3			<0.01			<0.01
Disease-free	54.1	25.3		92.3	45.6	
Recurrence	24.3	23.2		7.7	27.9	
Recurrence -> Died	10.8	40.4		0.0	17.7	
Died	10.8	11.1		0.0	8.8	
Stage 4			0.08			0.20
Disease-free	60.0	38.8		84.2	62.5	
Recurrence	0.0	4.1		0.0	5.0	
Recurrence -> Died	13.3	2.0		0.0	0.0	
Died	26.7	55.1		15.8	32.5	
All stages			<0.01			<0.01
Disease-free	67.5	46.2		90.5	65.1	
Recurrence	14.5	13.2		4.1	14.8	
Recurrence -> Died	8.4	20.8		1.3	7.1	
Died	9.6	19.8		4.1	13.0	

*p*-value was calculated with Fisher’s exact test or χ^2^-test as appropriated.

## 4. Discussion

This work aimed to explore whether managing MDT patients with ovarian cancer, including from the beginning in a diagnostic-therapeutic process, can impact recurrence and mortality 1 year and 2 years after diagnosis. 

Ovarian cancers represent a very complex category of neoplasms because they have different morphologies and behaviors and are characterized by a relatively late diagnosis due to the lack of specific symptoms [8,9]. The availability of new drugs [33,34] slightly improved the outcome of the disease. The use of PARP (Poli ADP-ribose polymerase) inhibitors also is effective on secondary and tertiary recurrences when the genetic background of the tumor allows it [35]. It was demonstrated that a collegial discussion of ovarian cancer could lead all the specialists to evaluate the diagnostic-therapeutic areas beyond their competence, increasing awareness of the number of potential treatments available and expected pitfalls and thus improving the effectiveness of treatments [36,37]. There are already previous works in the literature documenting how a multidisciplinary approach combined with more sophisticated diagnostic tools and new surgical techniques results in a decrease in morbidity and recurrence of breast cancer [15,38], with an increase in disease-free survival [16]. It is also interesting to note that the MDT is the key to good long-term control of the recurrence, even with curative intentions, since loco-regional recurrences represent more a marker that predicts the onset of distant metastases than a determinant of prognosis [17]. A multidisciplinary approach is strongly recommended for tumors of the stomach [39], colon [18], and certainly rectum [40], even if to review only a previous clinical decision [41]. The combined approach in advanced rectal cancers can reduce local recurrences and probably raise survival rates [18] and ensure acceptable toxicity [42]. In lung cancer, the MDT allows 80% of patients without disease in the first year, compared with 62.3% of those treated with standard methods [19]. Mortality at 10 years also shows that patients followed by the MDT have significantly lower mortality rates compared to the traditional approach both in the first year (OR = 0.68; 95%CI 0.51- 0.90) and in the first 3 years (OR = 0.5; 95%CI 0.36–0.7), with lower recurrence rates in MDTs (OR = 0.51; 95%CI 0.32- 0.79) [19]. Additionally, rare tumors, characterized by high mortality, seem to ensure, with the MDT method, a better survival or at least an improvement in the patient’s quality of life: this applies to cholangiocarcinoma [24], to head and neck tumors [21,22] and neoplasms of the esophagus [23]. Concerning ovarian cancers, population screenings have already shown poor efficacy in the early diagnosis and prevention of these neoplasms [43]. Since recurrences are often incurable, controlling the symptoms and the patient’s quality of life are elements to be considered in managing these patients [44]. Burton’s study focuses on applying cytoreductive surgery for ovarian cancer recurrences. In a list of 616 patients, 20 treated with the MDT approach had a median survival of 42 months following the procedure and 5-year survival of 45% [45]. 

In our experience, 24.1% of women were followed by MDT, which was introduced in our province in late 2015. The MDT group usually meets in person once a week. During the COVID-19 pandemic, the group continued its activity, also experimenting with a discussion in online mode. This modality has contributed to significantly improving the participation of professionals in the discussion of clinical cases. The diagnostic–therapeutic decisions are consistent with the guidelines, and if a deviation occurs, the reasons in support of this are made explicit through an IT reporting of the collective discussion. Furthermore, to support the collegial discussion, an IT report was implemented, which informs patients and professionals of the result of the collegial decision. 

In our study, MDT patients present at an earlier stage of diagnosis and are more likely to undergo surgery and receive chemotherapy. In terms of being disease-free, MDT patients have an advantage within one year and two years of diagnosis. If the recurrence rate remains comparable in the first year between the two groups, a benefit is instead observed two years after the diagnosis in the MDT group. Similarly, there is a clear advantage in recurrences followed by death and overall deaths. The most exciting fact is that, with the same stage in the two groups, this advantage is more evident in stages III and IV, which represent 77% of tumors and are responsible for the high mortality of these neoplasms. Supposing the MDT cannot always modify the prognosis, this approach could at least assure better management of recurrences with more personalized care based on the patient’s characteristics [35]. Personalized care becomes a fundamental pillar for these patients: an important aspect is the emotional management of patients so that MDT must also manage the patients’ fear related to the disease and recurrence and ensure a good quality of life by monitoring the side effects related to drug toxicity [46]. The patient’s psychological state and attitude in daily activities are strongly influenced by the fear of recurrences [47]. Care for cancer survivors requires a multidisciplinary effort and a team approach. There is also a need to increase knowledge about survivors’ treatment and long-term survival complications [48]. The diagnosis of ovarian cancer is a highly stressful event: the treatments have a strong impact not only on the body but also on the emotional sphere of the patients because they upset the routine of these women [49].

A final consideration concerns the fact that most ovarian tumors are diagnosed in advanced stages: in 2020 (the most recent year available in our study), 70% of tumors were diagnosed in stage III–IV, not dissimilar to the 74% reported in other studies in less recent years [50]. Advanced stages are always associated with a high risk of death (RR 10.54; 95% CI 9.16; 12.13) [51] or worse survival (5-year survival of advanced forms is 31% according to SEER data) [52].

Among the strengths of this study, we underline that they are population data, not affected by selection bias. The data are of good quality (for the variables considered, the missing value are always negligible) and refer to recent years.

Among the limits, we must mention that the size of the cohort analyzed is minimal; thus, mismatching in the two groups examined cannot be excluded. Furthermore, the lack of information on the type of surgical approach could be an element of the recurrence rate. 

Evidently, this study does not intend to validate the efficacy of MDT but only testify how much, in advanced forms of ovarian cancer, a multidisciplinary approach seems to lead to more encouraging results.

## 5. Conclusions

Our study does not show significant changes in 20 years in terms of incidence and mortality. Ovarian cancer continues to represent a challenge for gynecological surgeons and oncologists: the absence of cancer screening and the ability of early diagnosis strongly affect the prognosis of these patients. However, an MDT who takes care of the woman from the first suspicion of the diagnosis, addressing surgery and therapy without neglecting psychological and social aspects, could favor the prognosis of this disease.

## Data Availability

The data presented in this study are available on request from the corresponding author. The data are not publicly available due to ethical and privacy issues, requests of data should be approved by the Ethic Committee after the presentation of a study protocol.

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
