# Peer review of "Ovarian Cancer in a Northern Italian Province and the Multidisciplinary Team"

_cancers, 2022, doi:10.3390/cancers15010299_

Round 1

Reviewer 1 Report

This manuscript described an observational study including 448 ovarian cancer cases registered in an Italian Cancer Registry between 2012-2020. It tried to present recent advantages of MTD regarding Disease Free Survival (DFS) and death of ovarian cancer patients. It looked quite brief and meaningful. However, there were some critical defects in study design and analytical methodology.

1.       What was MDT? Which treatment measures were applied?

2.       What was the criteria for the patient inclusion of MDT? Apparently, there were a huge differences in clinicopathological characteristics between MDT patients and non-MDT patients. The two groups were not comparable.

3.       It was right that a stratification analysis by stage was performed. However, there were so many other features different, such as Morphology, surgery, and chemotherapy, et al. which severely affected the prognosis and would confound the results. Therefore, it is hard to judge the results validated.

4.       A person/month based PFS and OS would be better to estimate the prognosis.

5.       Too many unnecessary tables; for example, Table 1 and Table 3 could be integrated together. The forms of the tables were not standard (too many lines, et al).

Author Response

We thank the reviewer for the comments, we hope that our corrections are compliant with his requests.

Best regards, 

Lucia Mangone

Reviewer 2 Report

Review Report for the Manuscript “Ovarian cancer in a northern Italian province and the multidisciplinary team

Rating the Manuscript

Originality/Novelty: Is the question original and well defined? Do the results provide an advance in current knowledge?

Yes, the authors have studied how a multidisciplinary team (MDT) approach can improve the prognosis in terms of recurrence and death of ovarian cancer patients. They have used data from Italian Cancer registry between 2012-2020.

Significance: Are the results interpreted appropriately? Are they significant? Are all conclusions justified and supported by the results? Are hypotheses and speculations carefully identified as such?

Yes, the results are interpreted well, however conclusion section could be improved.

Quality of Presentation: Is the article written in an appropriate way? Are the data and analyses presented appropriately? Are the highest standards for presentation of the results used?

Yes, the article is written well. Data representation could be improved.

Scientific Soundness: is the study correctly designed and technically sound? Are the analyses performed with the highest technical standards? Are the data robust enough to draw the conclusions? Are the methods, tools, software, and reagents described with sufficient details to allow another researcher to reproduce the results?

Yes, the data is robust enough to draw conclusions and the methods, tools and methods used in the data analysis are explained properly.

Interest to the Readers: Are the conclusions interesting for the readership of the Journal? Will the paper attract a wide readership, or be of interest only to a limited number of people? (Please see the Aims and Scope of the journal)

Yes, this would be a great article for the researchers in the cancer research field.

Overall Merit: Is there an overall benefit to publishing this work? Does the work provide an advance towards the current knowledge? Do the authors have addressed an important longstanding question with smart experiments?

Yes. This study provides an advancement to the current knowledge. 

English Level: Is the English language appropriate and understandable?

Yes, English language in the manuscript is appropriate and understandable. 

Overall Recommendation: Accept after Minor Revisions

Given below are the comments for each section of the manuscript.

Abstract

The abstract is written and summarizes the content of the manuscript.

Line 27: “Disease Free Survival 1-year post-diagnosis, recurrences, recurrences-deaths and deaths were 67.5%, 14.5%, 8.4% and 9.6%, respectively, better than the non-MDT group (46.2%, 13.2%, 20.8 % and 19.8%, respectively) (p <0.01).”

Mention the CI for the p value.

Introduction

Introduction is well written.

Line 35: “Ovarian cancer (OC) represents the eighth most frequently diagnosed tumor and the

seventh most lethal cancer in women leading to almost 185,000 deaths annually world-wide.”

The paper cited here is from 2018, can you find more recent statistics for ovarian cancer?

Line 44: “Accordingly, 5-years survival remains very low (42.7%) in Italy, even for cases diagnosed in recent years (2010-2014).”

Is this the survival rate when the cancer is diagnosed at stage IV? I think the authors could discuss more about the difficulty in early detection of ovarian cancer and how the survival rate changes with the stage of diagnosis.

Line 51: “Several studies and scientific societies have advocated the centralization of OC patients, and a multidisciplinary approach is essential both in the decision-making strategies and during surgical procedures, but unfortunately, this recommendation is still disregarded.”

In all the other places authors have used the term ovarian cancer, but here they say OC. Please be consistent. 

At the end of the introduction authors could mention what analysis methods are used in the study.

Materials and Methods:

Methods section is very well written with all the required information. 

Results

Results section could be improved. Here authors have just stated the same information that are included in the tables.

Line 100: “From 2012 to 2020, 448 ovarian cancers were collected (Table 1).”

Authors need to reword this sentence. They could say data/information of 448 ovarian cancer patients were collected. 

Line 115: “Stage III presents, after 1-year, higher percentages of DFS (54.1% vs 25.3%) and lower recurrence-death values (10.8% vs 40.4%) (p <0.01). The same results are observed 2 years after diagnosis in the MDT group: DFS 92.3% vs. 45.6% and recurrence 7.7% vs. 27.9%; similarly, fewer recurrence/deaths and deaths were observed (p <0.01).”

Line 120: “At 2 years the values become statistically significant in the MDT group both for DFS (84.2% vs 62.5%) and for mortality (15.8% vs 32.5%; p <0.20).”

When p values are mentioned please state the CI.

3.2 Figures and Tables

Representation of data and tables could be improved.

Figure 1: 

Figure caption needs to be more informative.

Table 1:

Age at diagnosis: What’s the reason for selecting <50, 50-60 and 65+ as groups. 

Why didn’t you group the data into more groups?  For example, <40, 40-50, 50-60,65-70 and 65+. 

Methods of diagnosis: Authors could briefly discuss about these methods in the introduction section. Also, it’s not very informative when the authors mention “Clinical/Instrumental”

Table 3: Please state the CI for p values.

Discussion

Some of the facts in the introduction are repeated in the discussion section.

Line 190 in Discussion: Furthermore, an MDT approach appears to be able to reduce recurrences in breast cancer, rectal cancer , or reduce lung cancer mortality. Meaningful results were also obtained on head and neck and esophagus tumors, as well as cholangiocarcinoma.

Line 54 in Introduction: A multidisciplinary approach is strongly recommended for tumors of the stomach, of the colon and certainly of the rectum, even if to review only a previous clinical decision.

In the discussion section authors need to focus more on their own results and discuss about their findings. But here the authors have mainly focused on already reported data.

Conclusion:

Conclusion could be improved. Authors need to support the conclusion based on the results they obtained from their study.

References:

Some of the references are more than 10 years old. It they don’t contain important information authors could replace these with new references.

References: 19,21,22,26,28,41,46 and 49

Also, some of the references contains the doi, and others don’t. Please be consistent.

Author Response

(The authors gave the same response as above.)

Round 2

Reviewer 1 Report

I don't like the Tables.  It should have been claimed that the results need RCT to be validated.  

Author Response

Dear Reviewer,

thank you for your comment we hope that our answer is exhaustive.

Sincerely,

Lucia Mangone
